# Longer durations of piperacillin/tazobactam treatment cause more prolonged alteration of colonization resistance in mice

Bryan S. Hausman[1], Claire E. Kaple[2], Jennifer L. Cadnum[1], Samir Memic 🄳[1], Naseer Sangwan[3,4], Curtis J. Donskey 🄳[4,5]*

1 Research Service, Louis Stokes Cleveland VA Medical Center, Cleveland, Ohio, United States of America, 2 Department of Molecular Biology and Microbiology, Case Western Reserve University School of Medicine, Cleveland, Ohio, United States of America, 3 Lerner Research Institute/Lerner College of Medicine of Case Western Reserve University, Cleveland, Ohio, United States of America, 4 Department of Medicine, Case Western Reserve University School of Medicine, Cleveland, Ohio, United States of America, 5 Geriatric Research, Education, and Clinical Center, Louis Stokes Cleveland VA Medical Center, Cleveland, Ohio, United States of America

* curtis.donskey@va.gov

## Abstract

Broad-spectrum antibiotics that are excreted into the intestinal tract disrupt microbiota that provide colonization resistance against healthcare-associated pathogens. Minimizing the duration of treatment is a core element of efforts to reduce the adverse effects of antibiotics, but limited information is available on whether this approach preserves colonization resistance. Here, we used a mouse model to examine the impact of 1, 3, 6, and 10-days of treatment with the broad-spectrum antibiotic piperacillin/tazobactam on colonization resistance against vancomycin-resistant *Enterococcus faecium* (VRE) and carbapenemase-producing *Klebsiella pneumoniae* at 6-, 10-, and 24-days post-treatment; colonization resistance against *Clostridioides difficile* was tested at 6 days post-treatment. Colonization resistance was not altered at these time points by 1 day of treatment, whereas 3, 6, and 10 days of treatment caused more prolonged microbiota disruption and altered colonization resistance extending to 6, 10, and 24 days, respectively. In contrast, 5 or 10 days of treatment with aztreonam, a narrow-spectrum antibiotic with no activity against anaerobes, did not alter colonization resistance. These findings provide support for efforts to minimize the duration of antibiotic therapy as longer durations of piperacillin/tazobactam treatment caused greater and more prolonged alteration of colonization resistance. Our results also highlight the potential for short courses of broad-spectrum antibiotics to alter colonization resistance and suggest that selection of antibiotics that cause less disruption of anaerobes may be useful to preserve colonization resistance.

**Data availability statement:** The sequence data for this study can be accessed through the open repository Zenodo at DOI 10.5281/zenodo.16415070. All relevant data are within the paper and its Supporting information files.

**Funding:** This work was supported by a grant from the US Department of Veterans Affairs as part of funding for VA Sequencing Collaborations United for Research and Epidemiology (SeqCURE), which in turn received funding from American Rescue Plan Act funds. The funders had no role in study design, data collection and analysis, decision to publish, or preparation of the manuscript.

**Competing interests:** C. J. D. has received research funding from Clorox, GOJO, and Ushio America unrelated to the current study. All other authors report no potential conflicts. This does not alter our adherence to PLOS ONE policies on sharing data and materials.

## Introduction

To address the adverse effects of antibiotics, there has been an increasing emphasis during the past 25 years on efforts to reduce the duration of therapy [1,2]. In the early 2000s, Singh and Yu et al. [3,4] and Rice [5] argued that limiting treatment duration was more realistic than asking physicians to withhold antibiotics when faced with uncertainty in caring for patients with suspected infections. In 2003, an evaluation of the appropriateness of antimicrobial use in an acute care hospital demonstrated that most unnecessary days of therapy were due to administration for longer than recommended durations [6]. Subsequently, many randomized trials have demonstrated that shorter courses of antibiotic therapy are as effective as longer courses [1,2]. Current antibiotic stewardship guidelines recommend interventions to reduce antibiotic therapy to the shortest effective duration [7].

A primary rationale for minimizing the duration of antimicrobial therapy is to reduce selective pressure contributing to emergence of antimicrobial resistance and *Clostridioides difficile* infection (CDI) [5,8–12]. Increasing durations and cumulative doses of antibiotic exposure may increase the risk for CDI [13], and some randomized trials have demonstrated increased emergence of resistance with longer versus shorter courses of treatment [3,14]. However, many trials of different antimicrobial durations have not been powered to assess resistance outcomes and/or have not included screening for emergence of resistance. Notably, in 2 sub-studies of randomized trials of 7 versus 14 days of antibiotic treatment, no differences were detected in abundance of antibiotic-resistance genes or microbiota species diversity [15,16]. Moreover, even short courses (i.e., single doses to 3 days) of antibiotics can cause profound alteration of the intestinal microbiota and prolonged disruption of colonization resistance with increased susceptibility to pathogen colonization and overgrowth [17–24]. It is not known if reducing the duration of broad-spectrum antibiotic treatment will substantially decrease the extent and duration of alteration of colonization resistance. Here, we used a mouse model to test the hypothesis that longer durations of treatment with broad-spectrum antibiotics result in greater alteration of the intestinal microbiota and more prolonged disruption of colonization resistance to healthcare-associated pathogens.

## Methods

### Ethics statement

The Animal Care Committee of the Cleveland VA Medical Center approved the experimental protocol (#1590938). Work was conducted in compliance with the recommendations in the Guide for the Care and Use of Laboratory Animals of the National Institutes of Health. Mice were monitored at least daily throughout the experiments (16 days for the *in vitro* evaluation of colonization resistance in cecal contents; 17 or 39 days for the *in vivo* mouse model evaluation of the impact of different piperacillin/tazobactam treatment durations on colonization resistance; 16 days for the in vivo mouse model evaluation of the impact of different durations of aztreonam on colonization resistance). Mice were monitored for weight loss, ruffled fur, poor appetite, decreased ambulation, and huddling behavior. Mice that displayed huddling behavior

or reduced mobility were weighed 1x daily. Weight loss of greater than 15% body weight would trigger euthanasia within 2 hours. Based on Animal Care Committee of the Cleveland VA Medical Center guidance, no analgesics were provided for subcutaneous injections of antibiotics or orogastric gavage of test organisms as these procedures cause only minimal and transient (<30 seconds) distress. Soft bedding and other enrichment devices were provided.

Death of the mice during the experiment due to antibiotic treatment or colonization with the pathogens was not considered a likely outcome or planned experimental outcome as mice did not develop overt signs of illness in previous mouse model studies involving use of the test antibiotics and test organisms [8,18,19,21–24]. In the current study, mice treated with antibiotics and/or colonized with the test pathogens did not develop any overt signs of illness during the period of monitoring prior to planned euthanization. No mice were found dead prior to planned euthanasia. All mice were euthanized at the planned experimental timepoints (16 days for the *in vitro* evaluation of colonization resistance in cecal contents; 17 or 39 days for the *in vivo* mouse model evaluation of the impact of different piperacillin/tazobactam treatment durations; 16 days for the *in vivo* mouse model evaluation of the impact of different durations of aztreonam).

## Mice

Female, 8–10 week old, Non-Swiss albino (CF-1®) mice (2–8 per group; 110 total) weighing ~30g (Inotiv, Indianapolis) were housed in individual cages with plastic filter tops during the experiments to prevent cross-contamination among animals. At the conclusion of treatment or the described sampling period, mice were euthanized by sodium pentobarbitol overdose followed by cervical dislocation.

## Pathogens studied

*Enterococcus faecium* C68 is a previously described clinical vanB vancomycin-resistant *Enterococcus faecium* (VRE) isolate [18]. The minimum inhibitory concentration (MIC) of piperacillin/tazobactam for C68 is 1,250 µg/mL [18]. *Klebsiella pneumoniae* VA367 (Kp VA367) is a clinical carbapenemase-producing isolate used in previous mouse model studies with piperacillin/tazobactam MIC of 128 µg/mL [24]. *C. difficile* VA-17 is a clinical ribotype 027 strain. The MIC of piperacillin/tazobactam for VA-17 is > 256 µg/mL [25].

## Antibiotics used for mouse experiments

Piperacillin/tazobactam was used as the primary test antibiotic because it is a broad-spectrum agent commonly used in hospitals. Piperacillin/tazobactam is excreted in bile and inhibits anaerobes, facultative gram-negative bacilli, and susceptible enterococci [8,18]. Mice received subcutaneous piperacillin/tazobactam (8.0 mg piperacillin/day) for 1, 3, 6, or 10 days or subcutaneous phosphate-buffered saline (PBS) for 1 day as control with the final day of treatment being the same for each group. We have previously demonstrated that piperacillin/tazobactam promotes overgrowth of VRE and *K. pneumoniae* in mice [8,18,19] and in colonized patients [8,10,26]. In mice treated with 1–3 days of piperacillin/tazobactam, colonization resistance to VRE and *C. difficile* was disrupted 1–5 days after discontinuation of treatment but showed evidence of re-establishment by 11–12 days after treatment [21–23].

Aztreonam, a narrow-spectrum antibiotic that was used for comparison, is a monobactam antibiotic with activity against aerobic and facultative Gram-negative bacteria but not against anaerobes or Gram-positive bacteria [27]. Mice (8 per group) received subcutaneous aztreonam (3.0 mg/day) for 5 or 10 days. Previously shown in mice, aztreonam treatment for 5 days suppressed indigenous *Escherichia coli*, but did not alter enterococci or anaerobes, and did not alter colonization resistance [2,19]. However, it is not known if longer durations of narrow-spectrum antibiotics alter colonization resistance.

The dose of both antibiotics was based on the daily dose recommended for human adults (in mg per kg of body weight) [18,19].

## Microbiology

Stool or cecal content samples were serially diluted in phosphate-buffered saline (PBS) and plated on selective media. *C. difficile* cultures were processed inside an anaerobic chamber. The selective medias for Kp VA367, VRE C68, and *C. difficile* VA-17 included MacConkey agar (Becton Dickinson, Sparks, MD) with 0.5 μg/mL of imipenem/cilastatin, Enterococcosel agar (Becton Dickinson) supplemented with 6 μg/mL of vancomycin, and *C. difficile* Brucella agar (CDBA) [28], respectively. After incubation for 48–72 hours, we calculated the colony-forming units (CFU) per gram of stool. The lower limit of detection was $\sim 2 \log_{10}$ CFU.

### *In vitro* evaluation of colonization resistance in cecal contents

Initial experiments were conducted using an *in vitro* model of colonization resistance which allows testing of multiple pathogens with a small number of mice [28–31]. A timeline for the experiment is shown in Fig 1. Mice (2 per group; 10 total) were treated with 1, 3, 6, or 10 days of piperacillin/tazobactam or PBS for 1 day as described previously. Six days after the last antibiotic dose, mice were euthanized as described and cecal contents were collected and transferred to an anaerobic chamber within 20 min. The duration of the experiment was 16 days prior to euthanasia. The cecal contents

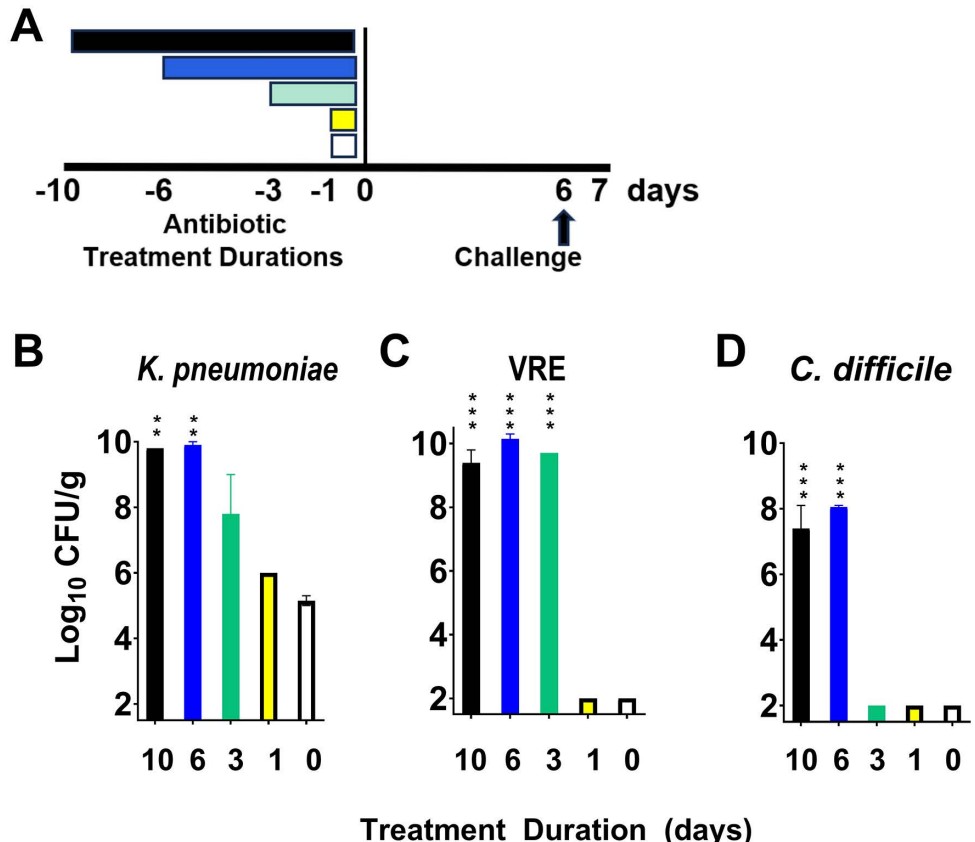

**Fig 1. Impact of different piperacillin/tazobactam treatment durations (timeline, A) on *in vitro* colonization resistance in cecal contents of mice (*n* = 2 per group; 10 total mice) inoculated with 10⁴ CFU of the *Klebsiella pneumoniae* VA367 (B), vancomycin-resistant *Enterococcus faecium* C68 (C), and *Clostridioides difficile* VA-17 (D) 6 days after the last antibiotic dose.** CFU, colony-forming units. Error bars show standard error.↑, challenge of cecal contents with test organisms. *=P<0.05, **=P<0.01, ***P<0.001.

were diluted threefold (volume/volume) in sterile, pre-reduced PBS. A final concentration of $10^4$ CFU/mL of the test organisms was added to aliquots of the cecal contents. The test organisms included Kp VA367, VRE C68, and *C. difficile* VA-17. The inoculated cecal contents were incubated anaerobically for 24 hours and then serial dilutions were plated on selective media.

### *In vivo* mouse model to assess the impact of different piperacillin/tazobactam treatment durations on colonization resistance 10 and 24 days after the final dose

Timelines for the experiments are shown with Figs 2–4. Mice (4–8 per group; 68 total) were treated with 1, 3, 6, or 10 days of piperacillin/tazobactam or PBS for 1 day as described previously. To assess *in vivo* colonization resistance to *K. pneumoniae* and VRE, separate groups of mice were challenged with $10^4$ CFU of Kp VA367 in combination with $10^4$ CFU of VRE C68 by orogastric gavage on days 10 or 24 after the final dose of piperacillin/tazobactam. These organisms were administered together to minimize the total number of mice used. Previous studies have demonstrated that VRE and *K. pneumoniae* occupy distinct niches and do not compete with one another in mice [32–34], and VRE and resistant gram-negative bacteria often co-colonize patients at high colonization densities [26]. To measure the concentration of the pathogens, stool samples from the mice challenged 10 days after the final dose were collected on days 11, 13, and 17 (1, 3 and 7 days after challenge) and stool samples from the mice challenged 24 days after the final dose were collected on days 25 and 29 (1 and 5 days after challenge). Mice were euthanized after the final stool collection. The duration of the experiment was 27 days for the mice challenged with the pathogens 10 days after the final antibiotic dose and 39 days for the mice challenged 24 days after the final antibiotic does.

### Effect of antibiotic treatment on the intestinal microbiota by culture and 16S rRNA amplicon sequencing

For the mice included in the experiments involving pathogen challenge at day 24, stool specimens were collected on days 0 (1 day after the final antibiotic dose for each group),1, 3, 7, 14, and 22. To assess the impact of piperacillin/tazobactam treatment on facultative Gram-negative bacilli and enterococci, serial dilutions of the stool in PBS were plated on MacConkey agar and Enterococcosel agar, respectively.

For the sequencing analysis, stool specimens (~120 mg) were collected on day 0 to provide a comparison of the effect of the different durations of piperacillin/tazobactam on the composition of the indigenous microbiota 1 day after the final dose and on day 14 to assess recovery of the microbiota 2 weeks after completion of treatment. Stool samples were stored at −80°C until processing. DNA Extraction and 16S sRNA amplicon sequencing were performed as described previously [34]. Briefly, DNA was isolated from stool samples using the QIAamp PowerFecal Pro DNA kit (Qiagen). The isolated microbial gDNA was checked for signs of degradation and quantified using the Bio-analyzer (Agilent) to ensure accurate sample input for the initial PCR step. A nested PCR method was used for amplification of the V4 region of the 16S rRNA gene and the addition of Illumina Nextera Unique Dual indexes. Afterwards, each library underwent standard quality control procedures to assess sample concentration and quality. Each library was pooled together ensuring equal sample distribution amongst sequencing reads. Amplicon sequencing was performed on an Illumina MiSeq with a 2x150 read length.

### Impact of different durations of the narrow-spectrum antibiotic aztreonam on *in vivo* colonization resistance

We examined the impact of 5- and 10-day courses of aztreonam on colonization resistance to VRE. Mice (8 per group; 32 total) received subcutaneous aztreonam (3.0 mg/day) for 5 or 10 days; stool specimens were collected on day 3 of aztreonam treatment to determine the impact on facultative Gram-negative bacilli. One day after the final aztreonam dose, $10^4$ CFU of VRE C68 was administered by orogastric gavage. Stool concentrations of VRE were measured 1, 3, and 6 days after gavage as described previously.

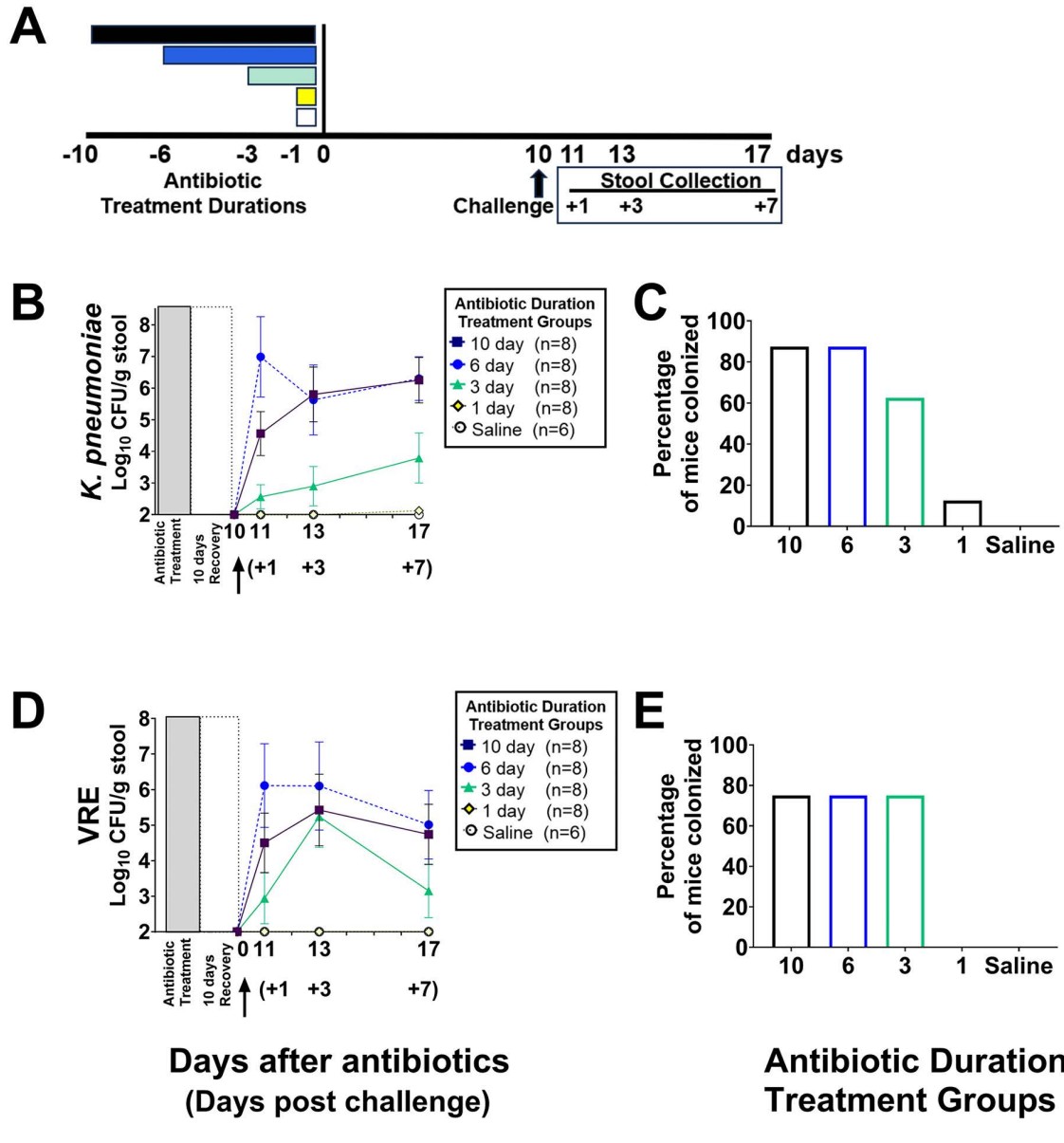

**Fig 2. Impact of the different piperacillin/tazobactam treatment durations (timeline, A) on *in vivo* colonization resistance to *Klebsiella pneumoniae* strain Kp VA367 (B) and vancomycin-resistant *Enterococcus faecium* (VRE) strain C68 (D) in mice (*n* = 38 total including 8 in the 10-, 6, 3, and 1-day treatment groups and 6 in the saline control group) challenged with the test organisms 10 days after the last antibiotic dose and the percentage of mice in each group with detectable colonization (C,E).** Data shown as means with error bars showing standard error. CFU, colony-forming units.↑, challenge with test organisms by orogastric gavage.

## Data analysis

Linear mixed models were used to compare concentrations of organisms among the treatment groups for the *in vivo* and *in vitro* mouse model experiments. Dunnett's method for multiple comparisons was used as a post-hoc test to compare the treatment groups against the saline controls. Data analysis was performed in R Version 4.2.2.

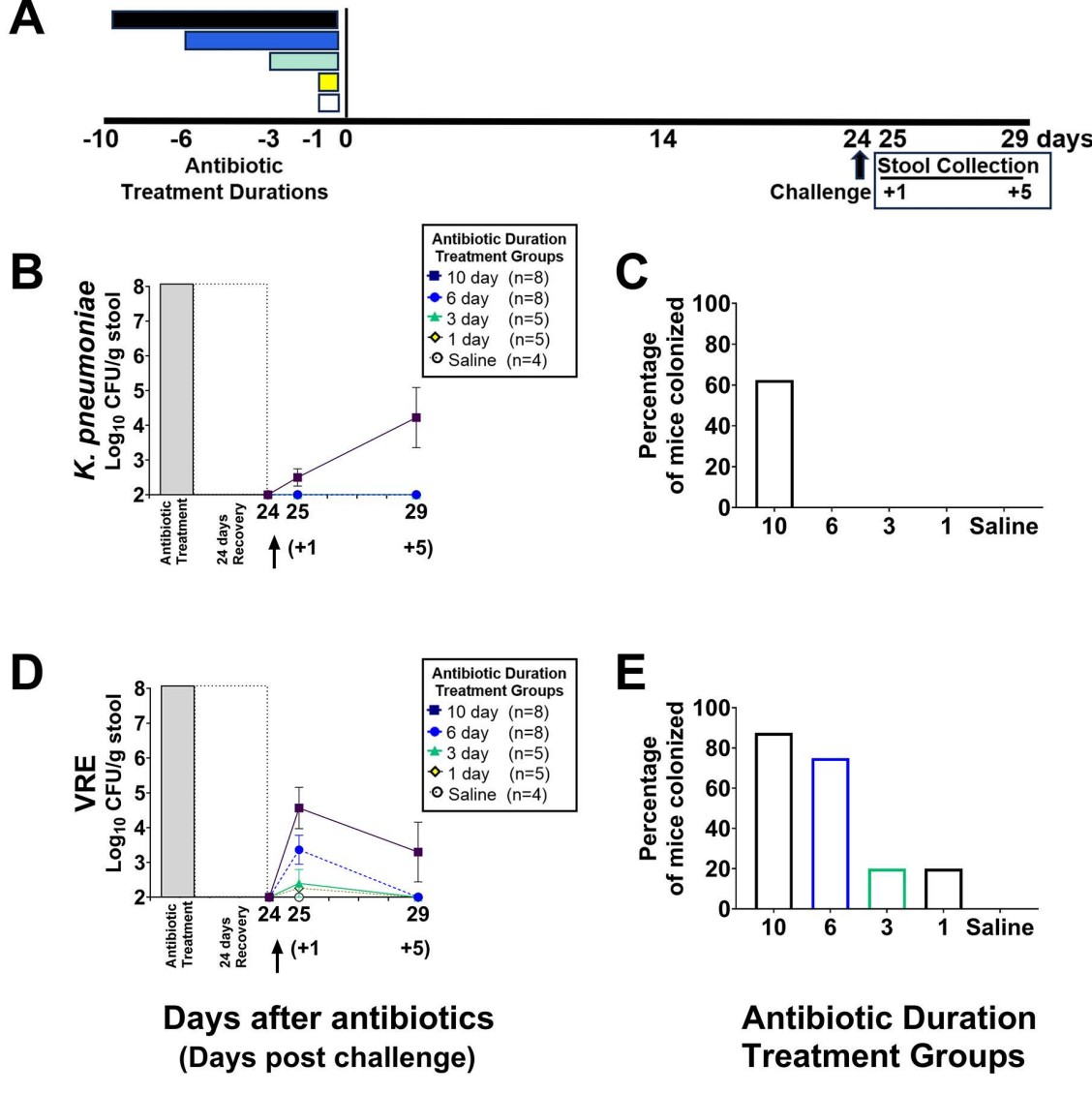

**Fig 3. Impact of the different piperacillin/tazobactam treatment durations (timeline, A) on *in vivo* colonization resistance to *Klebsiella pneumoniae* strain Kp VA367 (B) and vancomycin-resistant *Enterococcus faecium* (VRE) strain C68 (D) in mice (*n* = 30 total including 8 in the 10- and 6-day treatment groups, 5 in the 3- and 1-day treatment groups, and 4 in the saline control group) challenged with the test organisms 24 days after the last antibiotic dose and the percentage of mice in each group with detectable colonization (C,E).** Data shown as means with error bars showing standard error. CFU, colony-forming units. ↑, challenge with test organisms by orogastric gavage.

For analysis of the sequencing data, individual fastq files without non-biological nucleotides were processed using the Divisive Amplicon Denoising Algorithm (DADA) pipeline [35]. The output of the dada2 pipeline (feature table of amplicon sequence variants [an ASV table]) was processed for alpha and beta diversity analysis using phyloseq [36] and microbiomeSeq (http://www.github.com/umerijaz/microbiomeSeq) packages in R. Alpha diversity estimates were measured within group categories using estimate richness function of the phyloseq package. Canonical correspondence analysis (CCA) was performed using Bray-Curtis dissimilarity matrix between groups and visualized by using ggplot2 package [37]. Differential abundance analysis was performed using ANOVA in R software (The R Foundation for Statistical Computing,

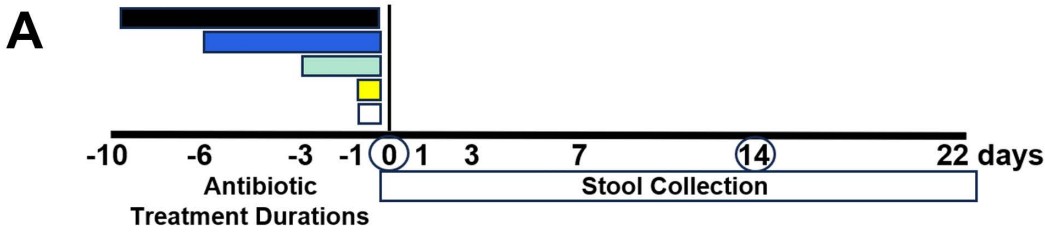

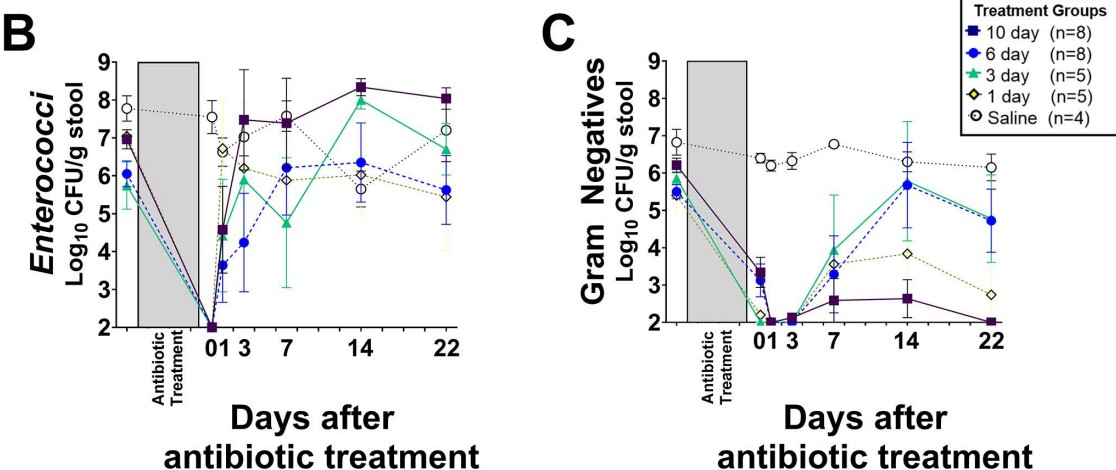

**Fig 4. Impact of different piperacillin/tazobactam treatment (timeline, A) on densities of indigenous enterococci (B) and facultative Gram-negative bacilli (C) in the same mice shown in Fig 3 (*n* = 30 total).** Data shown as means with error bars showing standard error. CFU, colony-forming units. Circled day 0 and 14 denotes 16S sample collection (Figs 5 and 6).

Vienna, Austria). As appropriate, we adjusted for multiple comparisons using the Benjamini-Hochberg False Discovery Rate (BH FDR) method while performing multiple testing on taxa abundance across groups [38]. Permutational multivariate analysis of variance (PERMANOVA) was performed on all coordinates obtained during CCA.

## Results

### Longer durations of piperacillin/tazobactam treatment cause greater alteration of *in vitro* colonization resistance when challenge occurs 6 days after the last treatment dose

Fig 1 shows the impact of different piperacillin/tazobactam treatment durations on *in vitro* colonization resistance (data and analysis shown in S1 Table) For Kp VA367, cecal contents of mice in the 6-, and 10-day treatment groups had significant growth in comparison to saline controls (P = 0.005), cecal contents of mice in the 3-day treatment group had increased growth that did not reach statistical significance (P = 0.053), and cecal contents of mice in the 1-day treatment group did not have significant growth in comparison to saline controls (P = 0.63). For VRE C68, cecal contents of mice in the 3-, 6-, and 10-day treatment groups had significant growth in comparison to controls (P < 0.001), whereas cecal contents of mice in the 1-day treatment group did not (P = 1.0); recovery of VRE was reduced in the cecal contents of the 1-day treatment group and the saline group in comparison to the initial inoculum of $10^4$ CFU/mL. For *C. difficile* VA-17, cecal contents of mice in the 6- and 10-day treatment groups had significant growth in comparison to the saline controls (P < 0.001), whereas cecal contents of mice in the 1- and 3-day treatment groups did not (P = 1.0); recovery of *C. difficile*

was reduced in the cecal contents of the 1- and 3-day treatment groups and the saline group in comparison to the initial inoculum of $10^4$ CFU/mL.

## Longer durations of piperacillin/tazobactam treatment cause more prolonged alteration of *in vivo* colonization resistance to VRE and *K. pneumoniae*

We next examined the impact of the different treatment durations on the recovery of colonization resistance to VRE C68 and Kp VA367 and the percentage of mice in each group with detectable colonization. We elected to challenge the mice at 10 and 24 days after completion of antibiotic treatment based on the *in vitro* results demonstrating alteration of colonization resistance at 6 days post-treatment (i.e., the goal was to evaluate colonization resistance at multiple time points post-treatment including the 24-day post-treatment challenge when recovery of colonization resistance was anticipated). When challenged 10 days after the last antibiotic dose (Fig 2 and S2 Table), the mice in the 6-day treatment group had significantly increased VRE colonization in comparison to saline controls (P = 0.011) and the mice in the 10-day treatment group had a non-significant trend toward increased colonization (P = 0.072); the 1- and 3-day treatment groups did not differ significantly from saline controls (P ≥ 0.41). When challenged 10 days after the last antibiotic dose, the mice in the 6- and 10-day treatment groups had significantly increased Kp VA367 colonization in comparison to saline controls (P ≤ 0.001), whereas the 1- and 3-day treatment groups did not (P > 0.70). For both test organisms, the number of mice with detectable colonization was lower in the 1-day treatment group in comparison to the 3-, 6- and 10-day groups.

When challenged 24 days after the last antibiotic dose (Fig 3 and S3 Table), the mice in the 10-day treatment group had a trend toward increased VRE colonization in comparison to saline controls (P = 0.09), whereas the other groups did not differ significantly from saline controls (P = 1.0). When challenged 24 days after the last antibiotic dose, the mice in the 10-day treatment group had significantly increased Kp VA367 colonization in comparison to saline controls (P = 0.04), whereas the other groups did not (P = 1.0). For Kp VA367, 60% of mice in the 10-day treatment group had detectable colonization, versus none in the other groups. For VRE C68, ≥ 75% of mice in the 6- and 10-day treatment duration groups had detectable colonization, versus 20% of the 3- and 1-day duration groups and none of the saline controls.

## Piperacillin/tazobactam treatment suppresses indigenous enterococci and facultative Gram-negative bacilli

Fig 4 and S4 Table show the impact of piperacillin/tazobactam treatment on densities of indigenous enterococci and facultative Gram-negative bacilli. For all treatment durations, both enterococci and facultative Gram-negative bacilli were significantly decreased at day 0 and day 3 in comparison to saline controls (P < 0.05). Levels of enterococci recovered more rapidly than facultative Gram-negative bacilli in all groups. Gram-negative bacilli remained below 3 $\log_{10}$ CFU per g of stool through day 22 in the 10-day piperacillin/tazobactam group.

## Longer durations of piperacillin/tazobactam treatment cause greater alterations of the intestinal microbiota based on 16S rRNA amplicon sequencing

We selected 14 days post-treatment in addition to day 0 for the 16S rRNA amplicon sequencing analysis based on evidence of recovery of indigenous enterococci and facultative Gram-negative bacilli by 14 days post-treatment. Fig 5A shows the bacterial diversity in stool specimens at day 0 (1 day after the last antibiotic dose) and day 14 for the different piperacillin/tazobactam treatment durations. Alpha (Shannon diversity index) analysis of 16S rRNA gene amplicon sequencing data revealed reduced diversity in piperacillin/tazobactam treatment groups in comparison to saline controls at both time points. The reductions in diversity were statistically significant in comparison to controls for the 6- and 10-day groups at day 0 and in the 3-, 6-, and 10-day groups at day 14 (P < 0.01 for each comparison). At day 14, the 1-day treatment group did not differ significantly from saline controls but demonstrated increased diversity in comparison to the 6- and 10-day treatment groups (P < 0.05).

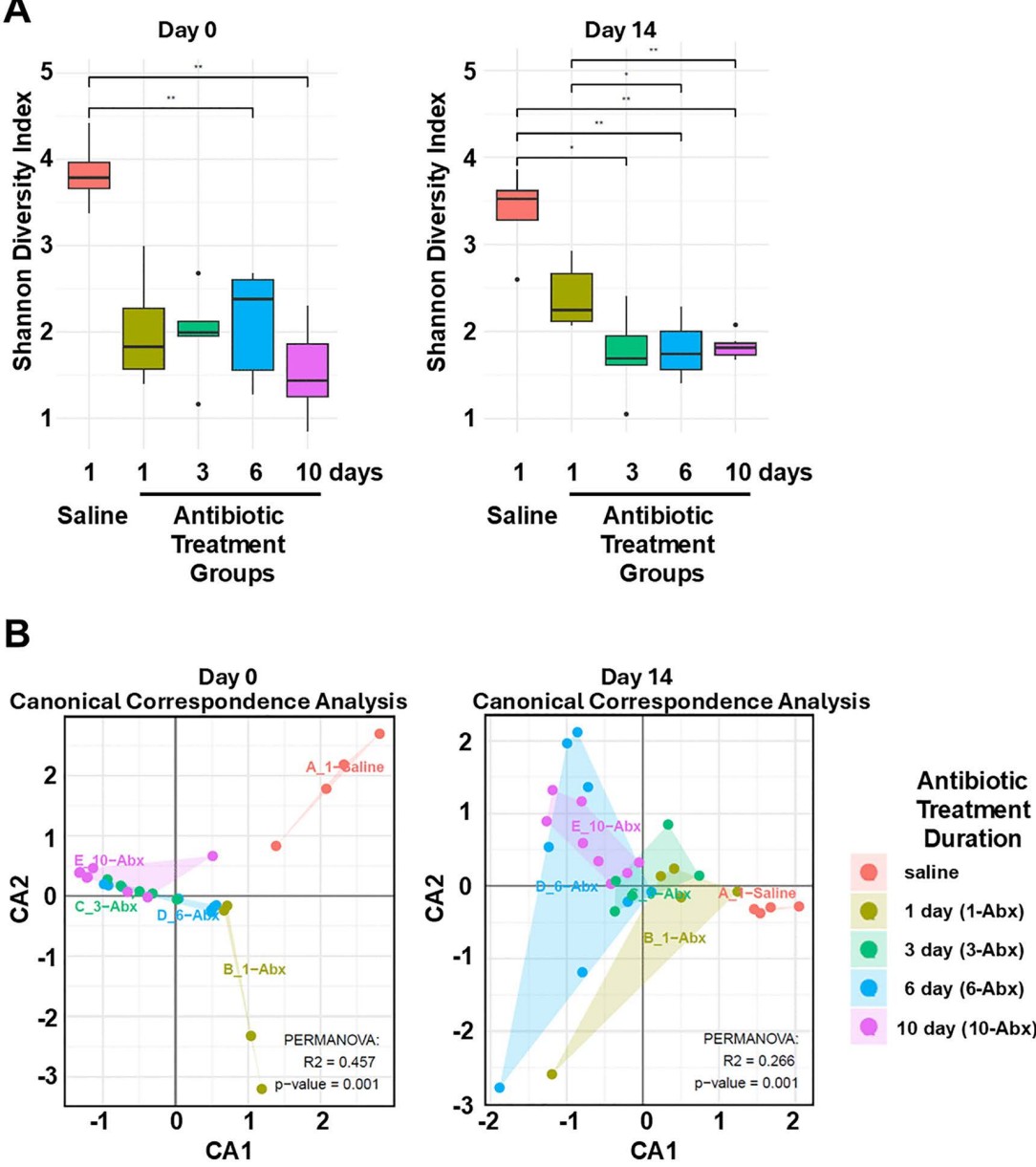

**Fig 5. Effect of subcutaneous antibiotic treatment on the intestinal microbiota in stool of the same mice shown in Fig 3 (*n* = 30 total) (time-line, Fig 4A) by 16S rRNA amplicon sequencing.** (A), alpha diversity in stool specimens at day 0 (1 day after the last antibiotic dose) and day 14 for the different piperacillin/tazobactam treatment durations; (B), beta diversity of the treatment groups by canonical correspondence analysis at day 0 and day 14. Black horizontal lines associated with the box plots represent median values. Box and whiskers represent interquartile range and minimum and maximum values, and black circles outside the whiskers represent outliers. CA, correspondence analysis. *, P<0.05; **, P<0.01.

Fig 5B shows the beta diversity of the treatment groups by canonical correspondence analysis at day 0 and day 14. In comparison to the control group, all piperacillin/tazobactam groups demonstrated a large treatment effect size (R2 = 0.457; P = 0.001) at day 0. At day 14, there was evidence of substantial recovery of the microbiota in the 1-day treatment group but not in the 3-, 6-, and 10-day treatment groups (R2 = 0.266; P = 0.001).

Fig 6A shows the relative abundance of 10 taxa at the genus level. At day 0, all the antibiotic duration groups had reduced relative abundance of *Lachnospiraceae* NK4A136 group in comparison to saline controls; the 3-, 6-, and 10-day treatment groups had substantial increases in the relative abundance of *Mycobacterium*, but the 1-day treatment group did not. At day 14, all the antibiotic duration groups had substantial recovery in the relative abundance of *Lachnospiraceae* NK4A136 group in comparison to saline controls and the increase in *Mycobacterium* was no longer present. Enterococcus had low relative abundance in all groups at day 0 consistent with the culture results showing suppression of enterococci by piperacillin/tazobactam during treatment; the relative abundance of enterococci was increased in the 3-day treatment group versus the 1-day group and was substantially increased in the 6- and 10-day treatment groups at day 14 after antibiotic treatment. Fig 6B shows the Differential abundance analysis (P < 0.05, ANOVA, Benjamini-Hochberg [BH]) for the 5 taxa at the genus level whose abundance was differentially altered the most by piperacillin/tazobactam treatment.

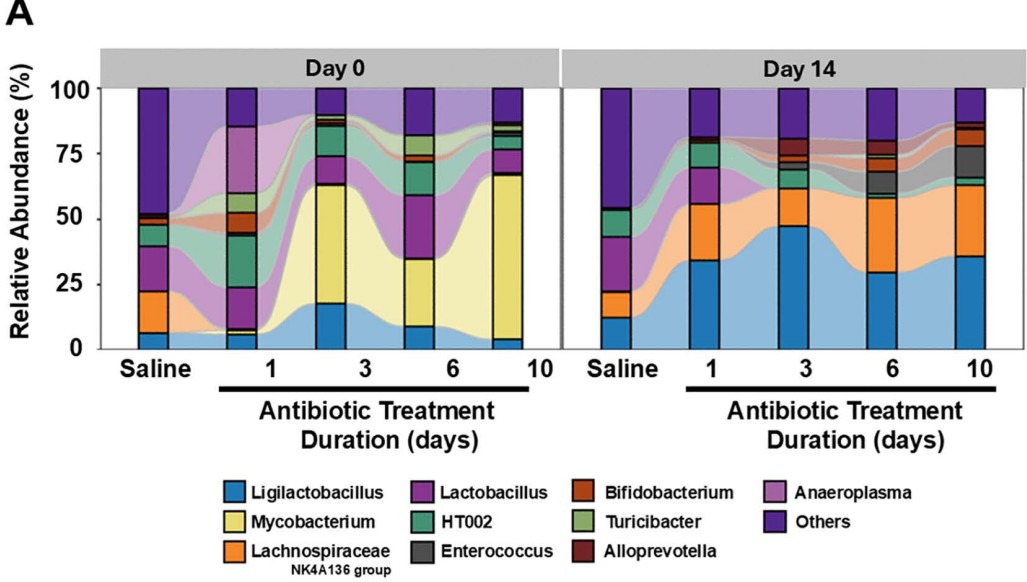

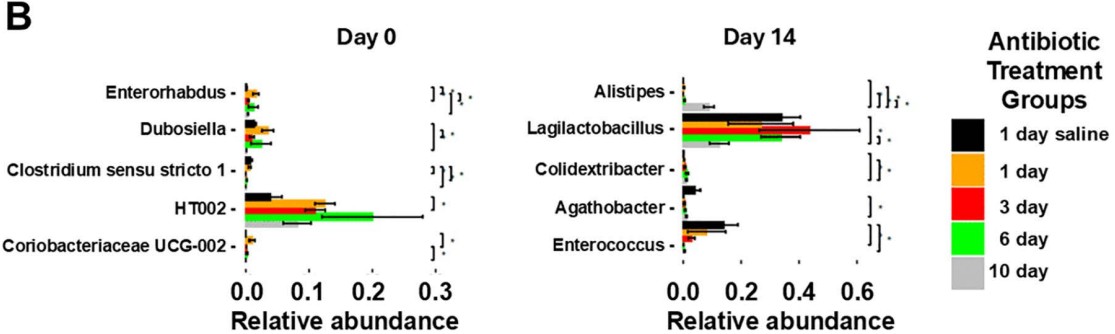

**Fig 6. (A) Relative abundance of 10 taxa at the genus level by 16S rRNA amplicon sequencing at day 0 (1 day after the last antibiotic dose) and day 14 for different piperacillin/tazobactam treatment durations in the same mice shown in** Fig 3 (*n* = 30 total) (timeline, Fig 4A); **(B) differential abundance analysis (P < 0.05, ANOVA, Benjamini-Hochberg highlighting the 5 taxa at the genus level whose abundance was differentially altered the most by piperacillin/tazobactam treatment.** 1 day, 3 day, 6 day, and 10 day indicate days of piperacillin/tazobactam treatment. *, P < 0.05; **, P < 0.01.

## Longer durations of the narrow-spectrum antibiotic aztreonam do not alter *in vivo* colonization resistance to VRE

Aztreonam resulted in suppression of indigenous facultative Gram-negative bacilli to undetectable levels on day 3 of treatment (data not shown). In comparison to saline controls, the 5- and 10-day aztreonam treatment courses did not promote colonization by VRE (P ≥ 0.77 for all comparisons) (Fig 7).

## Discussion

Minimizing the duration of therapy is a core element of efforts to reduce overuse of antimicrobials. However, relatively little information has been available on the impact of different durations of antibiotic treatment on intestinal colonization by healthcare-associated pathogens. Here, we demonstrated using a mouse model that longer durations of piperacillin/tazobactam treatment cause greater and/or more prolonged alteration of *in vivo* and *in vitro* colonization resistance to multiple pathogens, including VRE, *K. pneumoniae*, and *C. difficile*. For mice receiving 10 days of treatment, colonization resistance to *K. pneumoniae* remained significantly altered 24 days after the last antibiotic dose. These findings provide support for efforts to minimize the duration of antibiotic therapy.

Although our results suggest that reducing the duration of treatment may be beneficial, they underscore the fact that even short courses of broad-spectrum antibiotics can substantially impact colonization resistance. Piperacillin/tazobactam

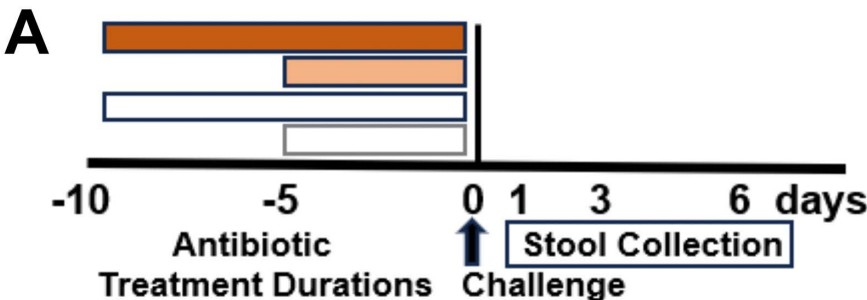

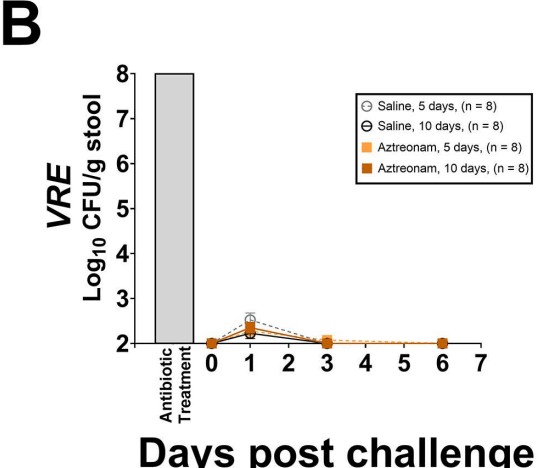

**Fig 7. Impact of the different aztreonam treatment durations (timeline, A) on *in vivo* colonization resistance to vancomycin-resistant *Enterococcus faecium* (VRE) strain C68 (B) in mice (*n* = 32 mice; 8 per group) challenged with the test organisms 1 day after the last antibiotic dose.** Data shown as means with error bars showing standard error. CFU, colony-forming units. ↑, challenge with test organisms by orogastric gavage.

treatment for 3 days altered colonization resistance to VRE and *K. pneumoniae* when challenge occurred 6 days after the last antibiotic dose. Although 1 day of treatment did not alter colonization resistance when challenge occurred 6–24 days after treatment, 1–2 days of piperacillin/tazobactam treatment has been shown to disrupt colonization resistance when challenge occurs soon after treatment (1–3 days) [18,19]. In clinical practice, efforts to reduce the duration of therapy to less than 3 days would be challenging and in most clinical trials examining different durations of therapy the shorter treatment durations have been 3 or more days [1,2].

The results of the sequencing analysis are consistent with previous evidence that piperacillin/tazobactam causes substantial and prolonged alteration of the intestinal microbiome [19,21]. The findings provide some evidence that longer durations of piperacillin/tazobactam treatment may cause greater disruption of the microbiome. In comparison to the controls, bacterial diversity was significantly reduced in stool of mice in the 6- and 10-day treatment groups at the end of treatment (day 0) and in the 3-, 6-, and 10-day treatment groups at day 14 post-treatment. There was evidence of substantial recovery of the microbiota in the stool of mice in the 1-day treatment group but not the other groups by day 14 after the last antibiotic dose.

In addition to minimizing the duration of therapy, efforts to select antibiotics that cause less disruption of intestinal anaerobes could be considered as an alternative strategy to reduce adverse effects on colonization resistance [8,10,34,39,40]. Many studies have demonstrated that antibiotics that have limited impact on the anaerobic microbiota have a lower propensity to promote overgrowth with healthcare-associated pathogens than antibiotics that suppress anaerobes [8,10,34]. In the current study, we confirmed that a relatively long (10-day) course of aztreonam, a narrow-spectrum antibiotic with no activity against anaerobes, does not disrupt colonization resistance to VRE. In a previous study, we demonstrated that 19 days of aztreonam treatment did not promote persistence of colonization by VRE [41]. Although selection of agents that do not disrupt colonization resistance is a promising strategy [8,39,40], relatively few currently available agents have minimal impact on anaerobes and factors such as cost and other adverse effects must be considered when selecting treatments.

Our findings should be interpreted in the context of the following limitations. We used a mouse model in which healthy mice were dosed daily and only 1 broad-spectrum antibiotic was studied. Antibiotic excretion in the intestinal tract of mice and humans may differ. Given that mice metabolize drugs more rapidly than humans and may have more rapid clearance of antibiotics from the intestinal tract [18,19], it is plausible that short antibiotic courses in mice might translate to longer durations in humans. However, we have previously demonstrated that piperacillin/tazobactam levels in stool of mice dosed once daily are comparable to levels reported in patients [21,28]. We did not assess the impact of aztreonam on the anaerobic microbiota. However, previous studies have demonstrated that aztreonam does not inhibit anaerobes in the intestinal tract [8]. The challenge with pathogens occurred once at different times after discontinuation of antibiotic treatment. In clinical settings, it is anticipated that patients may be exposed to pathogens multiple times, including during therapy. Different time points were used for the assessments of the impact on the microbiome and the assessments of colonization resistance. Piperacillin/tazobactam achieves high concentrations in bile and may inhibit establishment of colonization by susceptible pathogens such as *C. difficile* and VRE when exposure occurs during treatment [8]. To address these limitations of mouse model studies, there is a need for studies to assess the impact of different durations of antibiotic treatment on colonization resistance in patients. One potential approach to facilitate such evaluations might be to incorporate the use of *in vitro* models of colonization resistance that examine pathogen growth in stool suspensions into randomized trials of different treatment durations [28,30]. Finally, we focused on colonization resistance and did not assess the impact of different treatment durations on the emergence and persistence of resistance genes during therapy [12].

## Conclusion

We demonstrated using a mouse model that longer durations of piperacillin/tazobactam treatment cause greater and/or more prolonged alteration of *in vivo* and *in vitro* colonization resistance to multiple healthcare-associated pathogens. Our

results support efforts to reduce the duration of therapy but also highlight the potential for substantial alteration of colonization resistance even with short durations of broad-spectrum antibiotic treatment.

## Supporting information

**S1 Table. In vitro growth in cecal contents.**
(DOCX)

**S2 Table. In vivo challenge day 10.**
(PDF)

**S3 Table. In vivo challenge day 24.**
(PDF)

**S4 Table. Impact on GNB and enterococci by culture.**
(PDF)

## Acknowledgments

The authors thank the staff at the Cleveland VA Animal Facility.

## Author contributions

**Conceptualization:** Curtis J. Donskey.

**Data curation:** Bryan S. Hausman, Claire E. Kaple, Jennifer L. Cadnum.

**Formal analysis:** Samir Memic, Naseer Sangwan.

**Funding acquisition:** Curtis J. Donskey.

**Investigation:** Bryan S. Hausman, Claire E. Kaple, Jennifer L. Cadnum, Naseer Sangwan.

**Methodology:** Curtis J. Donskey.

**Resources:** Naseer Sangwan.

**Supervision:** Jennifer L. Cadnum, Curtis J. Donskey.

**Validation:** Curtis J. Donskey.

**Visualization:** Bryan S. Hausman.

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
