## [Decision Letter · Decision Letter 0]

22 Apr 2026

PONE-D-26-09003Longer Durations of Piperacillin/Tazobactam Treatment Cause More Prolonged Alteration of Colonization Resistance in MicePLOS One

Dear Dr. Donskey,

Thank you for submitting your manuscript to PLOS ONE. After careful consideration, we feel that it has merit but does not fully meet PLOS ONE’s publication criteria as it currently stands. Therefore, we invite you to submit a revised version of the manuscript that addresses the points raised during the review process. Your manuscript has been reviewed by two experts and a minor revision is needed before a final decision can be made. 

Please submit your revised manuscript by Jun 06 2026 11:59PM. If you will need more time than this to complete your revisions, please reply to this message or contact the journal office at plosone@plos.org. 

Your manuscript has been reviewed by two experts and a minor revision is suggested. Please follow the comments and do all necessary revision.

We look forward to receiving your revised manuscript.

Kind regards,

Yung-Fu Chang

Academic Editor

PLOS One

Journal Requirements:

“This work was supported by a grant from the US Department of Veterans Affairs as part of funding for VA Sequencing Collaborations United for Research and Epidemiology (SeqCURE), which in turn received funding from American Rescue Plan Act funds.”

“This work was supported by a grant from the US Department of Veterans Affairs as part of funding for VA Sequencing Collaborations United for Research and Epidemiology (SeqCURE), which in turn received funding from American Rescue Plan Act funds.”

“This work was supported by a grant from the US Department of Veterans Affairs as part of funding for VA Sequencing Collaborations United for Research and Epidemiology (SeqCURE), which in turn received funding from American Rescue Plan Act funds.”

“C. J. D. has received research funding from Clorox, GOJO, and Ushio America unrelated to the current study. All other authors report no potential conflicts.”

We note that you received funding from a commercial source: “Clorox”, “GOJO” and “Ushio America”

Reviewers' comments:

Reviewer's Responses to Questions

**Comments to the Author**

1. Is the manuscript technically sound, and do the data support the conclusions?

Reviewer #1: Yes

Reviewer #2: Yes

2. Has the statistical analysis been performed appropriately and rigorously? 

Reviewer #1: Yes

Reviewer #2: Yes

3. Have the authors made all data underlying the findings in their manuscript fully available?

Reviewer #1: Yes

Reviewer #2: Yes

4. Is the manuscript presented in an intelligible fashion and written in standard English?

Reviewer #1: Yes

Reviewer #2: Yes

5. Review Comments to the Author

Reviewer #1: Thank you for asking me to review this important manuscript (PONE-D-26-09003)

Title is accurate and informative

Abstract is clear and concise no issues

Introduction prepares to the study well and the aim is clear

The methods are well explained and detailed enough

Results The text is clear. The figures are very informative and detailed

However I do not see the footnote commenting on the figure below it. I see them in the text which is difficult to navigate. -Also the figures are NOT labelled clearly very tiny font on the top R hard to see (there are no numbers in the figures only in the text)

-I do not think there is enough explanation to the findings in the figures. I would hope that the authors would clearly highlight either in a footnote or on the figure the most important finding in the busy figures.

The discussion and conclusion are clear to the point, based on the data. just perfect and clear

The references are very reasonable

Reviewer #2: Using a mouse model, the authors examined how the duration of broad-spectrum antibiotic treatment affects colonization resistance. These extensive datasets serve as a foundation for future research aimed at reducing the negative consequences of antibiotic use. However, the experiments are not cohesively integrated. The changes of indigenous enterococci and gram-negative bacteria at 0, 1, 3, 7, 14, and 22 days post-treatment were presented in Figure 4. Results from one experiment should provide data that inform, refine, and necessitate the next experiments. Data presented in Figures 5 and 6 illustrate the impact of piperacillin/tazobactam treatment on the gut microbiome 0 and 14 days post-treatment. The effects of antibiotics treatment on the gut microbiota's ability to prevent pathogen invasion was assessed using cecal contents obtained 6 days post treatment. Yet, pathogen challenge (VRE and Klebsiella pneumoniae) was conducted at 10 and 24 days post-treatment. Despite the lack of data regarding microbiota changes or in vitro colonization resistance at those time points.

Translating these findings into clinical practice requires careful consideration of the interspecies differences between humans and mice. Because mice have a much faster metabolism and higher drug clearance rates than humans, a 3-day treatment in mice might translate to a 7- to 14-day human course. However, it is not well understood how antibiotic treatment effects on gut microbiota and colonization resistance differ between humans and mice. A more thorough discussion regarding the translation of study findings into clinical practice is required.

Supplemental data is unlinked to the text.

Minor points:

Lines 34-36; Conclusion of this study is not clear.

Lines 63-65; Please clarify what has not been determined yet, even though there are many reports that describe short courses of antibiotics still can disrupt colonization resistance [17-24]. Is it about the amount of disruption, the effects of broad-spectrum antibiotics, or something else?

Lines 80-82, lines 86-89 are not necessary information for the manuscript. Because there are no adverse events during the study (line 89-92).

Line 107, What is the MIC of C68? 1250 ug/mL or >1250 ug/ml? Please check.

Line 109, carbapenemase-producing Klebsiella pneumoniae is generally resistant to piperacillin/tazobactam. So, MIC of Kp VA367 would be >128 ug/ml. Please check.

Line 199; where are the data from day 3 stool specimens?

Lines 228-233; Please correct the subject word. The ‘mice’ were not correct subject for these sentences.

Line 341; The findings at Day 14 (3-day treatment group) are inconsistent with the data shown in Figure 4(B). Please discuss.

Lines 351-353: These pieces of information are not in Figure 6.

Lines 356-358; Are there any data to show the gut microbiota after aztreonam treatment?

6. PLOS authors have the option to publish the peer review history of their article (what does this mean?). If published, this will include your full peer review and any attached files.

Reviewer #1: **Yes:** Mohamed Yassin

Reviewer #2: **Yes:** Hosoon Choi

---

## [Author Response · Author response to Decision Letter 1]

28 Apr 2026

PONE-D-26-09003

Longer Durations of Piperacillin/Tazobactam Treatment Cause More Prolonged Alteration of Colonization Resistance in Mice

PLOS One

Response to Reviewers' comments:

Reviewer #1: Thank you for asking me to review this important manuscript (PONE-D-26-09003)

Title is accurate and informative

Abstract is clear and concise no issues

Introduction prepares to the study well and the aim is clear

The methods are well explained and detailed enough

Results The text is clear. The figures are very informative and detailed

However I do not see the footnote commenting on the figure below it. I see them in the text which is difficult to navigate. -Also the figures are NOT labelled clearly very tiny font on the top R hard to see (there are no numbers in the figures only in the text)

-I do not think there is enough explanation to the findings in the figures. I would hope that the authors would clearly highlight either in a footnote or on the figure the most important finding in the busy figures.

Response: The legends for the figures are included in the text as recommended in the instructions for manuscript preparation. Our understanding of the reviewer’s comment is that the reviewer found it difficult to navigate back and forth between the legends in the text and the figures when completing the review. This issue will be resolved when the figures are inserted into the text. We agree with the reviewer’s concern about the very small font in the legends for some of the figures (i.e., showing the antibiotic duration treatment groups and the numbers of mice). We have increased the font size and have used the recommended PLOS ONE tool to ensure that the figures are formatted in accordance with the journal requirements.

The discussion and conclusion are clear to the point, based on the data. just perfect and clear

The references are very reasonable

Response: We appreciate the comment.

Reviewer #2: Using a mouse model, the authors examined how the duration of broad-spectrum antibiotic treatment affects colonization resistance. These extensive datasets serve as a foundation for future research aimed at reducing the negative consequences of antibiotic use. However, the experiments are not cohesively integrated. The changes of indigenous enterococci and gram-negative bacteria at 0, 1, 3, 7, 14, and 22 days post-treatment were presented in Figure 4. Results from one experiment should provide data that inform, refine, and necessitate the next experiments. Data presented in Figures 5 and 6 illustrate the impact of piperacillin/tazobactam treatment on the gut microbiome 0 and 14 days post-treatment. The effects of antibiotics treatment on the gut microbiota's ability to prevent pathogen invasion was assessed using cecal contents obtained 6 days post treatment. Yet, pathogen challenge (VRE and Klebsiella pneumoniae) was conducted at 10 and 24 days post-treatment. Despite the lack of data regarding microbiota changes or in vitro colonization resistance at those time points.

Response: We agree that it may have been more ideal to harmonize the time points for all experiments and have noted this as a limitation of the study (lines 431 to 433). However, we view this as a minor limitation. We appreciate the comment that the results from one experiment should provide data that inform, refine, and necessitate the next experiments. We have clarified that we did use the results to inform subsequent experiments. We have noted that we selected 14 days post-treatment in addition to day 0 for the 16S rRNA amplicon sequencing analysis based on evidence of recovery of indigenous enterococci and facultative Gram-negative bacilli by 14 days post-treatment (lines 319 to 321). Our goal was to assess the impact of different antibiotic durations on colonization resistance at multiple time points. We initially assessed colonization resistance against multiple organisms by challenging cecal contents at 6 days post-treatment; based on the results, we selected 10 and 24 days post-treatment for the in vivo challenge and we selected 2 of the 3 organisms to test as the cecal content experiment demonstrated similar findings for all 3 organisms. We have noted that we elected to challenge the mice at 10 and 24 days after completion of antibiotic treatment based on the in vitro results demonstrating alteration of colonization resistance at 6 days post-treatment (lines 260 to 264).

Translating these findings into clinical practice requires careful consideration of the interspecies differences between humans and mice. Because mice have a much faster metabolism and higher drug clearance rates than humans, a 3-day treatment in mice might translate to a 7- to 14-day human course. However, it is not well understood how antibiotic treatment effects on gut microbiota and colonization resistance differ between humans and mice. A more thorough discussion regarding the translation of study findings into clinical practice is required.

Response: We agree and have noted as a limitation that mice metabolize drugs more rapidly than humans and may have more rapid clearance of antibiotics from the intestinal tract such that short durations in mice might translate to longer durations in humans (lines 422 to 425). We have noted that there is a need for studies in patients (lines 435 to 437).

Supplemental data is unlinked to the text.

Response: We have uploaded supplemental material.

Minor points:

Lines 34-36; Conclusion of this study is not clear.

Response: We have edited the abstract to make the conclusions more clear (lines 37 to 42).

Lines 63-65; Please clarify what has not been determined yet, even though there are many reports that describe short courses of antibiotics still can disrupt colonization resistance [17-24]. Is it about the amount of disruption, the effects of broad-spectrum antibiotics, or something else?

Response: We have clarified that it is not known if reducing the duration of treatment will reduce both the extent and the duration of alteration of colonization resistance (line 70 to 71). Although it has been demonstrated that short courses of antibiotics can disrupt colonization resistance, it has been proposed/hypothesized that minimizing the duration of treatment will be effective in reducing alteration of colonization resistance and emergence of resistance (lines 49 to 51 and 58 to 60).

Lines 80-82, lines 86-89 are not necessary information for the manuscript. Because there are no adverse events during the study (line 89-92).

Response: We added these comments based on input from PLOS ONE staff after our initial submission. It is correct that we did not anticipate any adverse effects of colonization based on our previous experience, but it is okay to leave this information if required as we did assess the health of the mice during the study and followed the noted procedures (lines 85 to 88 and 92 to 95).

Line 107, What is the MIC of C68? 1250 ug/mL or >1250 ug/ml? Please check.

Response: We have confirmed that that the MIC of piperacillin/tazobactam for VRE C68 is 1,250 mcg/mL (line 113).

Line 109, carbapenemase-producing Klebsiella pneumoniae is generally resistant to piperacillin/tazobactam. So, MIC of Kp VA367 would be >128 ug/ml. Please check.

Response: We have confirmed that the MIC for VA367 is 128 mcg/mL and have provided a reference (lines 113 to 115). The cut-off for piperacillin/tazobactam resistance in Enterobacterales is 32 mcg/mL (https://www.fda.gov/drugs/development-resources/fda-rationale-piperacillin-tazobactam-breakpoints-enterobacterales).

Line 199; where are the data from day 3 stool specimens?

Response: We collected stool on day 3 of aztreonam treatment to assess impact on indigenous facultative Gram-negative bacilli (lines 205 to 206). We have noted in the results that aztreonam resulted in suppression of indigenous facultative Gram-negative bacilli to undetectable levels on day 3 of treatment (data not shown) (lines 369 to 370); we elected to not include a figure for this data as we included this assessment primarily to confirm that aztreonam was being excreted into the GI tract and having the anticipated effect on indigenous Gram-negative bacilli.

Lines 228-233; Please correct the subject word. The ‘mice’ were not correct subject for these sentences.

Response: We have made this correction throughout the paragraph (lines 234 through 246).

Line 341; The findings at Day 14 (3-day treatment group) are inconsistent with the data shown in Figure 4(B). Please discuss.

Response: The culture results demonstrated increased concentrations of enterococci in the 3-day treatment group at day 14 (Figure 4B). We have clarified that the relative abundance of enterococci in the 3-day treatment group (Figure 6A) is increased versus the 1-day treatment group and substantially increased in the 6- and 10-day treatment groups at day 14 (lines 353 to 355). We agree that we might have expected a greater relative abundance of enterococci in the 3-day group in Figure 6A based on the culture results. However, as Figure 6A shows relative abundance and Figure 4B shows absolute abundance, it is likely that the discrepancy may be related to the abundance of other organisms in the gut microbiota that have increased to a greater degree than enterococci.

Lines 351-353: These pieces of information are not in Figure 6.

Response: We appreciate catching this error and have corrected the legend to remove the information that was incorrect (lines 364 to 365).

Lines 356-358; Are there any data to show the gut microbiota after aztreonam treatment?

Response: We have noted as a limitation that we did not assess the impact of aztreonam on the anaerobic microbiota (427 to 429).

---

## [Editor Report · Decision Letter 1]

7 May 2026

Longer Durations of Piperacillin/Tazobactam Treatment Cause More Prolonged Alteration of Colonization Resistance in Mice

PONE-D-26-09003R1

Dear Dr. Donskey,

We’re pleased to inform you that your manuscript has been judged scientifically suitable for publication and will be formally accepted for publication once it meets all outstanding technical requirements.

Kind regards,

Yung-Fu Chang

Academic Editor

PLOS One
---

## [Editor Report · Acceptance letter]

PONE-D-26-09003R1

PLOS One

Dear Dr. Donskey,

I'm pleased to inform you that your manuscript has been deemed suitable for publication in PLOS One. Congratulations! Your manuscript is now being handed over to our production team.

Kind regards,

on behalf of

Dr. Yung-Fu Chang

Academic Editor

PLOS One